# Kidney Transplantation in COVID Pandemic—A Review of Guidelines

**DOI:** 10.3390/jcm10132877

**Published:** 2021-06-29

**Authors:** Gabriela Gut, Agata Góral, Zofia Dal Canton, Paweł Poznański, Magdalena Krajewska, Mariusz Kusztal

**Affiliations:** 1Faculty of Medicine, Wroclaw Medical University, 50-556 Wroclaw, Poland; gabigut96@gmail.com (G.G.); agata.goral@student.umed.wroc.pl (A.G.); zofiadalc@gmail.com (Z.D.C.); 2Department of Nephrology and Transplantation Medicine, Wroclaw Medical University, Borowska 213, 50-556 Wroclaw, Poland; pawel.poznanski@umed.wroc.pl (P.P.); magdalena.krajewska@umed.wroc.pl (M.K.)

**Keywords:** COVID-19, renal transplantation guidelines, SARS-CoV-2, solid organ transplantation guidelines

## Abstract

The paper describes problems with the transplantation process during the COVID-19 pandemic. Transplantation procedures and programs have been impacted by COVID-19. The number of transplants has fallen noticeably. The first part of the paper points out changes in service organization, in particular donor and recipient pre-transplant and peri-transplant management. If the patients during pre-transplant evaluation need to attend face-to-face appointments, such as blood testing or other investigations, the risk of contracting or spreading COVID-19 should be minimized. “Clear green areas”, which are COVID-19-free pathways, are highly recommended in hospitals during transplant procedures. Diagnostic procedures concerning donors, including CT scans and coronavirus testing (nasopharyngeal swab), are necessary before transplant surgery. COVID-19 symptoms and risks of the transplant population are described. Detailed guidelines from transplant societies concerning changes in immunosuppression in infected recipients are discussed. Management of infected or suspected medical staff is mentioned. The paper ends with guidelines concerning vaccination against COVID-19 in transplant recipients.

## 1. Introduction

Coronavirus disease 2019 (COVID-19), caused by SARS-CoV-2, was first diagnosed during the outbreak in Wuhan, People’s Republic of China, in December 2019. The virus spread worldwide and on 11 March 2020, the World Health Organization declared a COVID-19 pandemic. The virus transmits between infected people through respiratory droplets and aerosols (breathing it in air, having it on eyes, nose, mouth or touching face with hands that have virus on them) [1]. According to data from the Center for Systems Science and Engineering at John Hopkins University, on 31 May 2021, more than 170 million positive cases have been confirmed with more than 3.55 million fatalities [2].

Transplant programs and all procedures related to transplantation have been impacted by the COVID-19 pandemic. Worldwide data concerning activities in organ donation and transplantation, derived from official sources, show that in 2020, the number of transplants from both deceased and living donors has fallen by 39% in Europe and by 35% in America, compared with 2019 (Table 1) [3,4]. Transplantations from deceased donors declined by 90% in France and by 51% in the USA. This mostly pertains to kidney transplantation, but the effect was also seen for heart, lung and liver, all of which provide a meaningful improvement in survival probability [5].

The challenge is to prevent the transmission of disease between patients and medical staff. Vaccination of the staff and/or patients can reduce the risk but does not fully guarantee safety against infection, especially with new mutated strains of SARS-Cov2.

Transplantation organizations have established various procedures for how to proceed before a planned transplantation. These recommendations refer to the types of tests and examinations that should be carried out on recipients and living donors, organization of hospitals, qualification process and the situations when healthcare workers should be tested. 

Recommendations pointed out that there is an opportunity to contact the clinic via online communicators or phone, to minimize the risk of infection, which would be increased during face-to-face contact.

The purpose of this review is policy analysis of guidance/recommendation of different societies for transplantation procedures during the COVID-19 pandemic. 

## 2. Methodology

A systematic search for guidelines for renal transplantation during the COVID-19 pandemic was performed from December 2020 to May 2021. We analysed recommendations with the latest publication date. Only English and Polish papers (position papers in journals and on websites) were reviewed. Guidelines from major organizations/societies were reviewed in terms of service organization, communication and risk management, transplant recipients and donors, qualification for transplantation, healthcare workers and treatment against COVID-19. Data were compiled in a table and compared for similarities and differences between the guidelines.

The following society/organizations addressed the problem: American Society of Transplantation (AST), American Society of Transplant Surgeons (ASTS), European Renal Association (ERA-EDTA), British Transplant Society (BTS), Canadian Society of Transplantation (CST), National Institute for Health and Care Excellence (NICE), Centers for Disease Control and Prevention (CDC), European Association of Urology (EAU) and Poltransplant (Polish Transplant Coordinating Centre).

## 3. Service Organization, Communication and Risk Management

The decision about limiting a transplant service or reopening it must consider patient wellbeing, prevalence of infection within the local community and the capability of a particular centre to manage the workload while maintaining infection control procedures and respecting recipient and donor autonomy [6].

The EAU guidelines recommend classification of kidney transplant patients into groups of priority for transplantation. Low priority patients are deferred if postponement by six months results in no clinical harm. Intermediate priority category needs case-by-case discussion. High priority and emergency category patients are defined as combined transplantation recipients or permanent lack/dysfunction of dialysis-access patients, respectively, and require renal transplantation urgently [7]. 

Transplant personnel should re-evaluate donors and patients on the waiting list. It is required to estimate the risks and benefits for them in the context of COVID-19. Decisions should be made by a multidisciplinary team [8]. It is crucial to ensure that there are COVID-19-secure sites or areas for pretransplant patients, donor assessment, donation and transplantation surgery, post-transplantation follow-up and ongoing care for all transplant recipients. Centres should be confident that they have the option of rapid turnaround testing for SARS-CoV-2 [8].

Poltransplant also recommends sectionalising “Clear green areas”, which are COVID-free pathways. Selection of living and deceased donors as well as organ harvesting procedures should be stopped in multi-specialized hospitals. Organ procurement can exceptionally be performed in those parts of a hospital where the area with non-infected COVID-19 patients is clearly defined and separated [9].

It is very important to minimize face-to-face contact with patients before and after transplantation. Before admission to hospital, there should be a medical interview, performed by means of distance communication, to ensure that patients have no COVID-19 signs or symptoms and that there is no evidence that patients should be quarantined [8].

For all stable patients with overall good general health and stable transplant function, visits to the hospital should be limited or postponed if possible. Contact with patients by telephone, video or email consultations is recommended whenever possible [7,8,9]. It is preferable to use different digital or remote methods to deliver prescriptions and medicine, for example, electronic prescription, pharmacy deliveries, postal services, drive-through medicine pick-up points. Most stable transplant outpatients are encouraged to take laboratory tests at home [5]. 

It is important to note that there are limitations of telehealth as well. Underserved patients without resources necessary for the effective use of telemedicine, such as access to the internet, computers, phones, may experience difficulties. Telemedicine may be less accessible also for racial/ethnic minorities, patients living in rural areas, with low literacy or low income [10].

If patients or potential living donors need to attend face-to-face appointments, such as blood testing or other investigations, the risk of contracting or spreading COVID-19 should be minimized. This can be done by avoiding any assistance accompanying a patient to medical appointments or, if it is really required, reducing the number of accompanying people to one. The use of public transportation should be avoided as well [8]. Reducing waiting times before medical appointments is crucial; it requires high discipline from patients (on time arrival) and medical services (right planning) [8].

It is important that transplant patients contact the Infectious Disease Department in case they show symptoms of COVID-19. Those patients should not reach the transplant centre [10].

Once all preparatory arrangements are complete, transplant recipients should be transferred to the closest transplant centre as soon as possible to limit migration of recipients [11].

## 4. Transplant Recipients

The most common symptoms of COVID-19 infection in transplant recipients are fever, cough and dyspnoea. Administration of immunosuppressive drugs leads to increased atypical or diminished presentation and, usually, gastrointestinal symptoms. Initial cohort report from the US epicentre suggests that patients with severe presentation are older and more likely to present with hypertension. However, the type of organ transplantation and disease severity is unrelated to the severity of symptoms [12]. The European Renal Association COVID-19 Database (ERACODA), settled by ERA-EDTA data, confirmed a high mortality rate among kidney transplant recipients with COVID-19 across Europe. Mortality due to COVID-19 among renal transplant recipients was 19.9% (17.5–22.5%) in 1013 patients [13].

In order to lower the risk of SARS-CoV-2 infection, the recipients must be treated as extremely vulnerable for COVID-19 development. That includes less frequent blood tests for routine monitoring, minimising hospital visits and using telephone or video consultations instead, adherence to general COVID-19 guidance (social distancing, hand hygiene, telework, travel restrictions) [7,8,9].

The ideal management of immunosuppression in transplant recipients with COVID-19 also remains largely unsettled. Most guidelines suggest continuing using usual quantities of immunosuppression according to established protocols [14]. However, CST guidelines suggest that decrease in immunosuppression should be considered, BTS guidance for all patients recommends stopping antiproliferative agents and considering calcineurin inhibitors reduction [15,16]. Manipulation of immunosuppressive drugs is a very subtle issue. These drugs may increase the risk of infection, lower the virologic control and change the clinical picture. Clinicians have extrapolated practices applied in case of other viral infections. In severe infection, they would lower the dose of antiproliferative drugs and calcineurin inhibitors. No significant conclusion can be drawn from these actions [12,15]. It is important to emphasise that this approach can increase the risk of systemic inflammation and graft rejection. Further data are essential given the lack of evidence-based treatment paradigms. 

According to the NICE guidelines, if organ recipients develop mild symptoms and do not require admission, a high dose of steroids is not recommended [15]. Increasing dose of hydrocortisone can lead to increased viral load. Returning to normal doses of immunosuppression can be considered 14 days after the onset of symptoms if the recipient is symptom-free in the absence of antipyretics for at least 3 days. For symptomatic patients requiring hospitalization, in addition to antiproliferative agents discontinuation and calcineurin inhibitors decrease or discontinuation, replacement of a standard steroid with dexamethasone is recommended. Administration of antiviral agents and adjunctive therapies should be considered. Blood oxygen saturation level should be maintained in a range between 92% and 96%. Because of increased probability of abnormal clotting parameters, D-dimer, prothrombin and aPTT (activated partial tromboplastin time) monitoring should be performed. Early thrombosis events prophylaxis with low molecular weight heparins is recommended. Patients that require ventilatory support should not receive antiproliferative agents, and calcineurin inhibitors should be reduced dramatically or stopped. High doses of steroids should be considered [15].

AST recommends having a 90-day medication supply at home in case of quarantine [17].

## 5. Organ Donors

The safety of living organ donors is just as important as the safety of transplant recipients. Medical history and contacts with people with COVID-19 over the last 28 days should be collected [7]. Social-distancing, masking and hand-hygiene measures must be respected for 14 days before transplant to avoid infection. Living donor candidates and members of their household may need to self-isolate for 14 days before peritransplant hospitalization. It is not mandatory, and a shared decision should be made based on the risk of contracting SARS-CoV-2, possibility of home office and local prevalence of COVID-19. However, self-isolating from the day of nasopharyngeal swab until admission is necessary [8,17].

Using organs from living donors with active COVID-19 is not recommended [9,17]. Transplantation should be postponed for at least 28 days with no symptoms and until the donor has a negative nasopharyngeal swab RT-PCR test result for SARS-CoV-2. For example, Poltransplant suggests that 28 days should be counted from the end of isolation [9]. If a donor does not present symptoms and initial COVID-19 infection occurred between 21 to 90 days before donation, transplantation can be performed, regardless of the test result. However, regarding the possibility of reinfection, if more than 90 days have passed since infection, a positive RT-PCR test should disqualify the donor. Renal donors should be additionally evaluated because of renal dysfunction associated with SARS-CoV-2 infection. If the donor is asymptomatic, transplant procedure should be delayed [8]. If RT-PCR test cannot be performed, Polish guidelines allow transplantation but only if epidemiological and clinical interview is negative and there are no COVID-19 manifestations in lung computed tomography (CT) images [9].

There are no clinical data that would accurately define the risk of transplant transmission of SARS-CoV-2. However, precautions are highly recommended. All potential deceased organ donors in the UK must have a negative SARS-CoV-2 RT-PCR test result. A positive result excludes donation [18]. Viral testing should be performed, at the earliest, 3 days before the procedure. Thoracic organ donors should be tested both from the lower and upper respiratory tract. Apart from that, donors should be screened epidemiologically and clinically. Antibody testing is not recommended, RT-PCR test remains the gold standard. Although antigenic tests are approved in some regions for use in potential donors, negative result must be verified by RT-PCR from a nasopharyngeal swab. For donors previously known to be COVID-19-positive, the recommendations are similar to living donors [18]. As the pandemic evolves, this approach may also change. In life-threatening situations, the risk of not finding a suitable non-infected donor is higher than the risk of transplanting an organ from a deceased donor with COVID-19 [19].

## 6. Qualification for Transplantation

During COVID-19 pandemic surges, it is necessary to maintain infection-control procedures [20]. Donors and recipients should be surveyed through an adequate history of exposure, symptoms of infection and recent hospitalization. Recommendations are saying clearly that all potential deceased and living donors and each potential organ recipient should be screened for epidemiologic and clinical history [9]. There is some variation in the recommended tests chosen to screen recipients in different centres around the world, but the most common is the nasopharyngeal swab RT-PCR test. Other tests are lung and intestine grafts and chest CT scans [14].

Most of the guidelines instruct to use RT-PCR tests in each potential donor not earlier than 72 h before transplantation [8]. Whether all human tissues may transmit the virus is unknown and debatable [14]. Some evidence suggests that transmission can be possible from lung, intestinal and possibly heart tissue, but not necessarily other tissues. Transfusion-related transmission has not been documented at this time, and blood donation agencies currently screen by acute symptoms only. Deceased infected tissues procured from infected deceased donors may pose a risk of transmission to the recipient as well as the transplant team, so if there is no certain evidence about virus transmission with particular organs, infection should be excluded in potential donors with no suspicion [14,17].

Deceased donors should be screened epidemiologically, including for known contacts, and by clinical history for suspected COVID-19. The AST suggests performing the first test within 3 days of procurement of deceased donor and the second one, as recommended by some experts, 24 h after the initial test. For living donors, the test should be performed, at the earliest, 3 days before organ transplantation [17].

Potential donors with history of COVID-19 are excluded from being organ donors for at least 28 days. Candidates should present no signs and symptoms and have a negative RT-PCR test to qualify for organ donation. The same rules concern organ recipients with COVID-19 history [8,9,16]. The Canadian Society of Transplantation requires two negative tests, at least 1 day apart [12]. Living donors and household members should be encouraged to self-isolate, particularly in the 14 days prior to donation, and to use preventive strategies, such as good hygiene and physical distancing [8,16].

Self-isolation for 14 days and negative test performed, at the earliest, 3 days before admission is also recommended for organ recipients [8]. RT-PCR test performed immediately prior to procedure is mandatory for all potential recipients [9].

Guideline concerns about performing chest CT scans routinely are diverse. NICE does not recommend it [8]. According to The American Society of Transplantation guideline, a CT scan of the chest cannot be relied upon neither to exclude nor diagnose SARS-CoV-2 infection in potential deceased or living donors and should not be used as the sole diagnostic modality [16]. Poltransplant suggests that a chest CT scan should be carried out for all donors with blood circulation. Potential donor with a negative RT-PCR test performed within the last 72 h and a negative chest CT scan can proceed with organ donation. When the RT-PCR test is negative but the chest CT scan is suspicious, the transplant team shall decide [9]. 

Organ procurement and transplantation are acceptable when the donor or recipient has not been tested, or the test result is yet unknown, but epidemiologic and clinical history are negative and chest CT scan does not show any changes characteristic for COVID-19 lung involvement. The director of the transplant centre decides after analysing the benefits and risks for the recipient. PCR test must be carried out directly before surgery, even though results will be available after procedure—it allows to start therapy early. Chest CT scan should be performed on all recipients directly prior to transplantation [9].

## 7. Healthcare Workers

The challenge of transplant programs is to minimize the risk of transmission between the staff and patients. 

In Italy, the RT-PCR test was applied to check only symptomatic patients, that is why an asymptomatic surgeon performed a kidney transplantation and was confirmed with COVID-19 after procedure. The physician did not present any of the common signs and symptoms. He has applied protection measures, such as hand hygiene, surgical mask and gloves. Both the patient and colleagues were examined and had negative tests. In conclusion, in times of pandemic, medical personnel should consider themselves as if they were potentially asymptomatic positive patients and use protective equipment [21]. To protect both the staff and the patients, guidelines suggest strict compliance with the epidemiological procedures and use of personal protective equipment such as gloves, eye protection, medical gown and N95 masks—or surgical masks, when supplies are limited. 

All guidelines suggest that medical staff with symptoms of COVID-19 should be prioritized for SARS-CoV-2 testing. Workers who live with a person with known or suspected COVID-19 should self-isolate and be tested as well. The CDC points out that the test can be considered also in case of asymptomatic healthcare providers with known or suspected exposure to SARS-CoV-2 and as a part of screening [22]. According to NICE, personnel who have been tested positive should self-isolate at least 10 days from symptoms onset or 10 days from the test with a positive result [8]. However, guidance written by Public Health England indicates that personnel who have been hospitalized because of COVID-19 should be isolated for 14 days [23].

Personnel who have a positive test result should be exempted from retesting within 90 days of the first test unless new symptoms occur. If the test is positive, check for common symptoms of COVID-19 to interpret the result. If, 90 days after a positive test, a person is tested and found positive, this should be considered as a new infection [24]. 

The CDC defines two strategies for returning to work: symptom-based strategy and test-based strategy. The test-based strategy is no longer recommended but, in some cases, could be considered as allowing healthcare workers to return to work earlier than the symptom-based strategy does. Test-based strategy allows symptomatic workers to return to work when the fever disappears without antipyretics, the symptoms improve and results from at least two respiratory specimens (PCR) collected at least 24 h apart are negative. For non-symptomatic medical staff, this strategy involves performing at least two tests 24 h apart, which must be negative to allow workers returning to work. The symptoms-based strategy considers the symptoms. Healthcare providers with mild to moderate illness who are not severely immunocompromised can return at least 10 days after the first symptoms appeared, at least 24 h have passed since the last fever occurred and symptoms have decreased. Workers with severe to critical illness or severely immunocompromised could return after at least 10–20 days since the first symptoms occurred and at least 24 h have passed since the last fever and when overall symptoms have decreased. This medical staff should consider a consultation with infection control experts. Asymptomatic healthcare workers can return 10 days after the first positive test if they are not severely immunocompromised. If they are severely immunocompromised, they can return to work at least 10–20 days since the first positive test occurrence [25]. 

Recommendation for general practitioners states that patients after transplantation should not have contact with other potentially infected patients in the waiting room. If such a person comes to the clinic, they should be separated and admitted as first priority, if possible. If the aim is to write a prescription, such a case does not require a visit to the clinic [9,26].

Summary of society/organization specific recommendations regarding transplantations during COVID-19 pandemic is displayed in Table 2.

## 8. Vaccine against COVID-19

Early release of information about vaccines to prevent COVID-19 is promising. Although neither of the available vaccines has been tested until March 2021 in transplant candidates or recipients, all societies and organizations focused on renal patients prioritize kidney patients, especially on dialysis maintenance, and kidney care professionals along with residents of long-term care facilities and health care workers, in receiving access to COVID-19 vaccines [10,14,25]. 

The mRNA vaccines from Pfizer and Moderna showed 94.1–95% efficacy in preventing COVID-19 infection in immunocompetent people. Vaccine efficacy in patients older than 65 years appears similar to the efficacy in younger patients. Data also suggest that when a breakthrough infection occurs, the disease is generally mild, showing the vaccines are also effective in preventing severe disease. Data regarding the durability of vaccine titres are still being gathered, although it currently appears that antibody titres persist for at least 4 months [27]. The current guidance is that everyone receives the vaccine, irrespective of past COVID19 infection or prior evidence of humoral immunity. There are case reports of immunosuppressed patients developing COVID-19 reinfection, suggesting lack of appropriate immune response or waning immunity after the first infection [28]. In the recent study about humoral response to two doses of an mRNA SARS-Cov-2 vaccine among 658 solid organ transplant recipients, 98 patients (15%) had detectable antibody response after two doses and 301 (46%) stayed without humoral response [29].

There are many indications that the adoption of the vaccine will change a lot in the peri-transplant procedure and allow transplantation programs to be reopened. Rules and recommendations for transplant candidates and recipients are summarized in Table 3 [30,31,32,33,34,35].

## 9. Recent Challenges

There is increasing population of post-COVID-19 patients on waiting lists. As some of them have experienced COVID pneumonia, the possibility of irreversible lung tissue damage is a serious question. In the case when chest CT or radiogram suggests fibrotic changes, a functional test with subsequent lung test and a pulmonologist’s opinion are expected.

Due to quicker lowering of post-vaccination antibodies post-transplantation, it is anticipated that, in case of COVID-19 vaccine, an additional shot (or schedule) will be required. Studies will give the answer at the end of 2021.

## 10. Conclusions

Medical professionals are facing challenges during this time because scientific evidence about COVID-19 infection is scarce, and the strategies must be based on expert opinions rather than evidence. Main guidelines present similar recommendations for transplant centres, organ donors and recipients and healthcare workers. Differences are minor.

Each of them suggests that reviewing precisely potential donors and recipients to estimate the risks and benefits in the context of COVID-19 is of primary importance.

It is crucial to minimise face-to-face contact with stable patients showing overall good general health.

Because of the lack of evidence-based research, guidelines about immunosuppressive drugs doses vary. Most guidelines also do not describe how to treat an organ recipient with active COVID-19. This can lead to arbitrary choices while administering drugs and other medical procedures.

Risk of SARS-CoV-2 transmission from deceased donors is also undetermined. All the guidelines suggest that positive donors should be excluded. However, for some patients, benefits of transplantation can outweigh the risks of infection.

Each guideline recommends that healthcare workers must use personal protective equipment to increase protection for both staff and the patient. In case of any COVID-19 symptoms, it is recommended to self-isolate immediately and perform an RT-PCR test. Determining when a medical staff can return to work depends on whether or not the symptoms are present, the overall state of health and the time which has passed since the test or symptoms have occurred.

## Figures and Tables

**Table 1 jcm-10-02877-t001:** Kidney transplant statistics in Europe, America and Poland for 2019 and 2020.

Region/Year	Total Kidney Transplants		Deceased Kidney Transplants		Living Kidney Transplants	
	2020	2019	2020	2019	2020	2019
Global	42,948	105,231	33,348	64,514	9264	40,720
America	25,582	39,515	19,515	28,035	6047	11,480
Europe	17,366	28,329	13,833	20,476	3217	7853
Poland	751	983	720	931	31	52

**Table 2 jcm-10-02877-t002:** Summary of society/organization specific recommendations regarding transplantation procedures during COVID-19 pandemic [7,8,9,10,11,12,13,14,15,16,17,18,19,20,21,22,23,24,25,26].

Service organization, communication and risk management	Minimising hospital visits, assurance of COVID-19-secure sites or areas EAU—classification of patients into groups of priority
Transplant recipients	EAU—continue to use standard immunosuppression according to established protocols CST—consider decrease in immunosuppression BTS—consider stopping administering antiproliferative agents and calcineurin inhibitors reduction NICE—high doses of steroids not recommended, antiproliferative agents discontinuation and calcineurin inhibitors decrease or discontinuation
Organ donors	28 days with no symptoms, negative RT-PCR test result NICE—if infection occurred 21–90 days before donation, transplantation can be performed, regardless of the test result
Qualification for transplantation	All potential deceased and living donors and each potential organ recipient should be screened with epidemiologic and clinical history Canadian Society of Transplantation—two negative test results NICE—CT not recommended American Society of Transplantation—CT not recommended Poltransplant—CT recommended
Healthcare workers	Strict compliance with the epidemiological procedures and use of personal protective equipment NICE—personnel should self-isolate at least 10 days Public Health England—personnel should self-isolate at least 14 days CDC—workers can return at least 10 days after the first symptoms appeared, at least 24 h have passed since the last fever occurred and symptoms have decreased
Treatment availability against COVID-19	CDC and EMA—Remdesivir if glomerular filtration rate <30 mL/min/1.73 m^2^

**Table 3 jcm-10-02877-t003:** Basic rules and timing of vaccination against COVID-19 in transplant recipients and candidates—summary based on references [30,31,32,33,34,35]. Abbreviations: aHUS, atypical haemolytic uremic syndrome; AR, acute rejection; ASTS, American Society of Transplant Surgeons; IS, immunosuppression; Tx, transplant.

(1)	Candidates for solid organ transplantation on a waiting list should be vaccinated before transplantation
(2)	Patients after Tx should be vaccinated if there are no strong contraindications
(3)	COVID-19 vaccination of patients early after Tx with standard IS regiment should be deferred for +/− 3–6 months after Tx (1 month in Poltransplant recommendation)
(4)	COVID-19 vaccination of patients early after Tx with IS regiment consisting of T cell ablative therapy should be deferred for +/− 3–6 months after Tx
(5)	After treatment of AR, vaccination should be deferred +/− 1 month
(6)	COVID-19 vaccination of patients after Tx who underwent B cell ablative therapy (RTX) should be deferred for +/− 3–6 months and B cell screen should be performed (count of B cells in peripheral blood sample)
(7)	mRNA vaccines are considered not to increase the risk of AR; however, there is lack of evidence of immunoresponsiveness for vaccination in patients after Tx
(8)	Patients with aHUS and kidney transplant on eculizumab should be vaccinated against COVID-19
(9)	There is lack of recommendations for vector based vaccines; however, the ASTS underlines that up to now, they have not recommended live viral vector vaccines for transplant patients

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
