# Peer review of "Kidney Transplantation in COVID Pandemic—A Review of Guidelines"

_jcm, 2021, doi:10.3390/jcm10132877_

Round 1
Reviewer 1 Report
Dear authors,
Thank you for the opportunity to review your manuscript.
The current paper reviews and compares the guidance/recommendations from transplantation organizations/societies for transplantation procedures during the COVID-19 pandemic. The review is timely and explores an important issue how transplantations societies and organizations are preparing and dealing with the transplantation practices during this current pandemic and its impact on overall transplantations. I have few comments/questions related to this current literature review.
- It is not clear from the objectives, if authors are conducting a narrative review or a focused literature review, or if it is a policy analysis of guidance/recommendation of different societies or organizations? This should be clearly stated in the manuscript.
- To make it explicit, authors can specifically mention in their objectives that they have reviewed the guidance/recommendation in terms of - Service organization, Communication and Risk management, eligibility of Transplant recipients, eligibility of organ donors, qualification for transplant, Qualification for Healthcare workers and Treatment availability against COVID-19 since, in fact they did.
- A parenthesis is missing for a reference 8 – line 179, page 4.
- To the benefit of readers, it would be great if authors can also include a table comparing/contrasting organizations/societies’ recommendation on service organization, communication and risk management, eligibility of transplant recipients, eligibility of organ donors, qualification for transplant, qualification for healthcare workers and treatment availability against COVID-19. Table 2 merely provides a summary of what was already stated in the conclusions and seems redundant here.
- Authors should also include the quality/grade of the evidence which was found a place in transplant societies/organizations’ recommendations/guidance or use a published quality checklist for grading the evidence for the same. This will be helpful to the readers to draw their own conclusions.
Author Response
Dear Reviewer,
We are very gratefully for deep insight and suggestions how to improve the manuscript. Please accept our responses and explanations given below. Thank you so much for your time and expertise.
#1 It is not clear from the objectives, if authors are conducting a narrative review or a focused literature review, or if it is a policy analysis of guidance/recommendation of different societies or organizations? This should be clearly stated in the manuscript.
response: Thank you for this remark. This is in fact policy analysis of guidance/recommendation of different societies. This has been included in introduction section.
#2 To make it explicit, authors can specifically mention in their objectives that they have reviewed the guidance/recommendation in terms of - Service organization, Communication and Risk management, eligibility of Transplant recipients, eligibility of organ donors, qualification for transplant, Qualification for Healthcare workers and Treatment availability against COVID-19 since, in fact they did.
response: brilliant remark. This sentence has been included
#3 A parenthesis is missing for a reference 8 – line 179, page 4.
response: It has been fixed
#4 To the benefit of readers, it would be great if authors can also include a table comparing/contrasting organizations/societies’ recommendation on service organization, communication and risk management, eligibility of transplant recipients, eligibility of organ donors, qualification for transplant, qualification for healthcare workers and treatment availability against COVID-19.
response: We appreciate for this remark. Table 2 has been produced and inserted in the revised version to fill this expectation.
#5 Authors should also include the quality/grade of the evidence which was found a place in transplant societies/organizations’ recommendations/guidance or use a published quality checklist for grading the evidence for the same. This will be helpful to the readers to draw their own conclusions.
response: We regret that could not use any reporting standard for grading/quality assesment like PRISMA for systematic review. In fact this is policy analysis of guidance/recommendation of different societies expressing opinions or suggestions rather then giving evidence. We hope readers identify this.
Reviewer 2 Report
The authors investigated the guideline review for COVID-19 in kidney transplantation. They focused on the collected data from many transplantation organizations or societies to compare recommendations and guidance for transplantation procedure during COVID-19 pandemic. They described that it is especially important to take the measures for COVID-19 pandemic in everyone involving in transplants including transplant recipients, organ donors, healthcare workers. They also stated the newly information about quantification of transplantation and vaccine against COVID-19. The Methods section is clear. In general, the article provides a useful review of recent studies on allograft kidney transplantation during COVID-19 pandemic.
Minor comments
I have the following concerns.
- The authors demonstrated about flu vaccination on page 5, line 230-234. I think the sentence is unnecessary because they described not influenza infection but COVID-19.
- The authors demonstrated “In some cases, for patients with life-threatening conditions, the risk of not finding another suitable, noninfected match is higher than organ transplantation from carefully selected deceased donors with COVID-19”. I think it is difficult to understand the meaning of this sentence. You should rewrite this easily and clearly.
- On table 1, several data were wrong, for example “4072” to “40720” in Global region of living kidney transplants, “1148” to “11480” in America of living kidney transplants. You should correct them.
- I note that the author described the reference number without brackets on page2, line 80 and on page4, line 179. You should correct them.
The manuscript is improved as a result of the revisions, and the authors have adequately addressed most of the comments from the original review. There is just one point that the authors have not properly or fully addressed, noted below.
Author Response
Dear Reviewer,
We are very gratefully for deep insight and suggestions how to improve the manuscript. Please accept our responses and explanations given below. Thank you so much for your time and expertise.
- The authors demonstrated about flu vaccination on page 5, line 230-234. I think the sentence is unnecessary because they described not influenza infection but COVID-19.
- Response: the sentence has been removed
- The authors demonstrated “In some cases, for patients with life-threatening conditions, the risk of not finding another suitable, noninfected match is higher than organ transplantation from carefully selected deceased donors with COVID-19”. I think it is difficult to understand the meaning of this sentence. You should rewrite this easily and clearly.
- response: thank you for this remark. The stantence is rewrited.
- On table 1, several data were wrong, for example “4072” to “40720” in Global region of living kidney transplants, “1148” to “11480” in America of living kidney transplants. You should correct them.
- I note that the author described the reference number without brackets on page2, line 80 and on page4, line 179. You should correct them.
- Response: Thank you for pointing out errors in manuscript. They are fixed in revised version.
Reviewer 3 Report
manuscript should be carefully re-read, and missing words/miswritten sentences should be corrected.
The review points out a few of the questions in transplant-medicine; however, my overall feeling is that the content is a bit to 'general'.
Maybe the reader is more supported when specifiek groups are emphasized
Author Response
Dear Editor and Reviewers,
We are very gratefully for deep insight and suggestions how to improve the manuscript
"manuscript should be carefully re-read, and missing words/miswritten sentences should be corrected."
Response: Following your suggestion many errors and sentences has been rewritten or corrected in revised version. Moreover the manuscript is enriched with new table and two references.
Reviewer 4 Report
The manuscript is well written including some interesting data from international transplant associations. However, recent publications from large national registries and data regarding immunity after vaccination in transplant recipients could be included. Additionally, a methodology section should be included in the manuscript.
Author Response
Dear Reviewer,
We are very gratefully for deep insight and suggestions how to improve the manuscript. Please accept our responses and explanations given below. Thank you so much for your time and expertise.
"recent publications from large national registries and data regarding immunity after vaccination in transplant recipients could be included.
response: Following your suggestion we added sentence with new reference: In the recent study about humoral response to 2 doses of mRNA SARS-Cov-2 vaccine among 658 solid organ transplant recipients, 98 patients (15%) had detectable antibody response after 2 doses and 301 (46%) stayed without humoral response[Boyarsky BJ, Werbel WA, Avery RK, Tobian AAR, Massie AB, Segev DL, Garonzik-Wang JM. Antibody Response to 2-Dose SARS-CoV-2 mRNA Vaccine Series in Solid Organ Transplant Recipients. JAMA. 2021,5:e217489.]
"Additionally, a methodology section should be included in the manuscript."
Response: We appreciate for this remark. New section was inserted with following description:
A systematic search for guidelines for renal transplantation during COVID-19 pandemic was performed from December 2020 to May 2021. We analyzed recommendations with the latest publication date. Only English and Polish papers (journal and website position papers) were reviewed. Guidelines from major organizations/societies were reviewed in terms of service organization, communication and risk management, transplant recipients and donors, qualification for transplantation, healthcare workers and treatment against COVID-19. Data were compiled in a table and compared for similarities and differences between the guidelines.
Reviewer 5 Report
Overall, I believe this is a good review about COVID-19 practices for our transplant patients. I like how the paper considers and summarizes the guidelines of many organizations, specifying certain guidelines. I have some minor suggestions below.
Line 34 - Please check to see if this still hold true, my understanding was that this was less likely the cause of transmission
Line 41 -change to 35-39% vs 39-35%
Line 47-I believe “stuff” should be the word staff
Line 54-55 -please change the sentence to ensure it flow better
Line 78- Transplant authorities should be changed to something such as transplant personnel
Line 93- nor should be or
Line 119 - should say present with hypertension
Line 179-there should be parenthesis around the reference number 8
Section 2 - I think it is important to discuss limitations of telehealth as well. For example, undeserved patient without access to the internet, computers or smart phones may have difficulty with this. Also, patient without modes of communications other than public transportation will also have a difficult time.
Section 7- Vaccine against COVID-19. I believe this seminal paper should be mentioned. The following paper describes the response of transplant pts the covid-19 vaccine
"Antibody Response to 2-Dose SARS-CoV-2 mRNA Vaccine Series in Solid Organ Transplant Recipients"
Author Response
Dear Reviewer,
We are very gratefully for deep insight and suggestions how to improve the manuscript. Please accept our responses and explanations given below. Thank you so much for your time and expertise
"Line 34 - Please check to see if this still hold true, my understanding was that this was less likely the cause of transmission
Response: We appreciate for this remark. You are right in recent statement surface transmissions are not so important - we rewrite the sentence
"Line 41 -change to 35-39% vs 39-35%
Line 47-I believe “stuff” should be the word staff
Line 54-55 -please change the sentence to ensure it flow better
Line 78- Transplant authorities should be changed to something such as transplant personnel
Line 93- nor should be or
Line 119 - should say present with hypertension
Line 179-there should be parenthesis around the reference number 8"
response: all pointed out errors/mistakes has been corrected in revised version of the manuscript
Section 2 - I think it is important to discuss limitations of telehealth as well. For example, undeserved patient without access to the internet, computers or smart phones may have difficulty with this. Also, patient without modes of communications other than public transportation will also have a difficult time.
response: Than you for this remark. We added following sentence with new reference:
It’s important to note that there are limitations of telehealth as well. Underserved patients without resources necessary for the effective use of telemedicine like access to the internet, computers, phones may have difficulty with this. Telemedicine may be less accessible also for racial/ethnic minorities, patients living in rural areas, with low literacy or low income [10].
Section 7- Vaccine against COVID-19. I believe this seminal paper should be mentioned. The following paper describes the response of transplant pts the covid-19 vaccine
"Antibody Response to 2-Dose SARS-CoV-2 mRNA Vaccine Series in Solid Organ Transplant Recipients"
response: This is great suggestion. We added this ref. and new sentence:
In the recent study about humoral response to 2 doses of mRNA SARS-Cov-2 vaccine among 658 solid organ transplant recipients, 98 patients (15%) had detectable antibody response after 2 doses and 301 (46%) stayed without humoral response[29].
Round 2
Reviewer 3 Report
The manuscript is more easy to read and understand now the corrections are made.
Author Response
I also see that all suggested by rewiever changes significantly improved the paper. Thank you again for that.
Reviewer 4 Report
No further comments.
Author Response
Suggested by rewiever changes significantly improved the paper. Thank you again for that.